# A scoping review of interventions to improve strength training participation

Jasmin K. Ma[1,2]*, Jennifer Leese[1,2], Stephanie Therrien[1], Alison M. Hoens[1,2,3], Karen Tsui[1], Linda C. Li[1,2]

1 Arthritis Research Canada, Richmond, British Columbia, Canada, 2 Department of Physical Therapy, University of British Columbia, Vancouver, British Columbia, Canada, 3 Centre for Health Evaluation and Outcome Sciences, University of British Columbia, Vancouver, British Columbia, Canada

* Jasmin.ma@ubc.ca

## Abstract

### Background

Low participation rates (1–31%) and unique barriers to strength training (e.g., specialized knowledge, equipment, perceived complexity) suggest effective strength training interventions may differ from effective aerobic or general physical activity interventions. The purpose of this scoping review was to examine interventions used to improve strength training participation through mapping theory, intervention characteristics, prescription parameters, and behaviour change techniques.

### Methods

Recommendations by Levac et al. (2010) and PRISMA-ScR were followed in the conduct and reporting of this review, respectively. Patients and exercise professionals participated in developing the research question and data extraction form, interpreting the findings, and drafting the manuscript. Medline, Embase, PsycINFO, CINAHL, SPORTDiscus, and PubMed databases (inception–December 2020) were searched. The inclusion criteria were (a) original peer-reviewed articles and grey literature, (b) intervention study design, and (c) behavioural interventions targeted towards improving strength training participation. Two reviewers performed data screening, extraction, and coding. The interventions were coded using the Behaviour Change Technique Taxonomy version 1. Data were synthesized using descriptive and frequency reporting.

### Results

Twenty-seven unique interventions met the inclusion criteria. Social cognitive theory (n = 9), the transtheoretical model (n = 4), and self-determination theory (n = 2) were the only behaviour change theories used. Almost all the interventions were delivered face-to-face (n = 25), with the majority delivered by an exercise specialist (n = 23) in community or home settings (n = 24), with high variability in exercise prescription parameters. Instructions on how to perform the behaviour, behavioural practice, graded tasks, goal setting, adding objects to the

**Data Availability Statement:** The dataset supporting the conclusions of this article is available in the supplementary files and in the Open Science Framework repository, https://osf.io/

dyxsq/?view_only=
12ff9dbc56dc46e4916d83fda21048e4.

**Funding:** JM is supported by the Michael Smith Foundation for Health Research Trainee Award (#17936), the Arthritis Society Post-Doctoral Fellowship (TPF-18-0209), and the Canadian Institute of Health Research Post-Doctoral Fellowship (201910MFE-430114-231890). LL is supported by the Harold Robinson/Arthritis Society Chair in Arthritic Diseases award, the Canada Research Chair Program, and the Michael Smith Foundation for Health Research. Funding bodies did not play a role in the study collection, analysis, interpretation of the data, or writing of the manuscript. Funder websites: https://cihr-irsc.gc. ca/e/193.html https://www.msfhr.org https:// arthritis.ca.

**Competing interests:** The authors have declared that no competing interests exist.

environment (e.g., providing equipment), and using a credible source (e.g., exercise specialist delivery) comprised the most common behaviour change techniques.

## Conclusions

Our results highlight gaps in theory, intervention delivery, exercise prescription parameters, and behaviour change techniques for future interventions to examine and improve our understanding of how to most effectively influence strength training participation.

## Introduction

International physical activity guidelines for public health recommend that adults and older adults should engage in muscle strengthening activities at least twice weekly [1–4]. Strength training involves the use of resistance (e.g., machines, body weight, resistance bands, free weights) to increase muscular strength. Regular strength training can provide benefits to cardiometabolic health such as lowering blood lipids [5], blood pressure [6], and risk for Type II diabetes [7, 8], and has been shown to reduce anxiety [9], depressive symptoms [10], and allcause mortality [11, 12]. It plays a crucial role in protecting older adults' continued independence and cognitive and physical functioning [13, 14]. The positive effects of strength training on multiple health outcomes across diverse populations underline the importance of treating the improvement of strength training participation rates as a valuable public health target.

Although the health benefits of strength training are known, population-level engagement is low, with only 1–31% of individuals (varying across countries as well as general and patient populations) meeting current strength training guidelines [15–20]. These rates are considerably lower than those for meeting the current aerobic exercise guidelines (~50%) [21, 22]. The low participation rates in strength training may be attributed to barriers that are observed in both aerobic and strength training (e.g., perceived lack of time, self-efficacy, cost [23, 24]) as well as barriers that are uniquely experienced when participating in strength training [25, 26]. Strength training is often perceived as complex, requiring specialized equipment and specific knowledge, and targeted at athletes [25, 27–29]. When compared to aerobic activities like walking, the perceived knowledge and effort required to effectively perform strength training (e.g., specific technique, how much strength training to do, how to progress, time needed, cost of equipment) can be discouraging [25, 27, 28]. Interventions targeting strength training may therefore require different behaviour change strategies than those targeting aerobic exercise or strength training combined with other physical activities.

Several reviews have synthesized general physical activity behaviour change interventions across diverse populations [30–32]; however, no synthesis of interventions specifically targeting strength training participation has been conducted. One systematic review summarized the behavioural, demographic, intrapersonal, interpersonal, and environmental factors associated with participating in strength training [25] and highlighted potential correlates—such as self-efficacy, intention, affective judgments, self-regulation, and subjective norms—linked to greater strength training behaviour. However, behaviour change interventions are complex, and the mode of delivery, providers, intervention dose, setting, behaviour change techniques (BCTs), and the exercise prescription itself may also influence the effectiveness of an intervention. BCTs, or the 'active' ingredients of an intervention [33], have been extensively summarized across health behaviours such as smoking, diet, and physical activity but have not been examined specifically across strength training interventions [30, 32, 34]. With so many

variables to consider when designing strength training participation interventions, it is important to summarize what has been tested to help set a research agenda for intervention development. The purpose of this scoping review was to map the intervention characteristics, prescription parameters, and BCTs used in interventions to improve strength training participation to date.

## Methodology

The PRISMA-ScR Checklist [35] was used to guide the reporting of this review (S1 File). We followed recommendations for the conduct of scoping reviews by Arksey and O'Malley (2005) with updated recommendations proposed by Levac et al. (2010) [36, 37].

**Availability of data and materials.** The dataset supporting the conclusions of this article is available in the Open Science Framework repository, https://osf.io/dyxsq/?view_only=12ff9dbc56dc46e4916d83fda21048e4. This protocol was registered at Prospero (https://www.crd.york.ac.uk/prospero/; Registration CRD42019120251).

## Search strategy

As per recommendations by Levac et al. (2010), the authors iteratively developed the literature search strategy, inclusion/exclusion criteria, and data extraction table, and met at the beginning, middle, and final stages of the review process to discuss challenges and insights and ultimately refine the methods. We consulted a medical librarian to develop the literature search strategy. We searched electronic databases for relevant articles and hand-searched trial registries and reference lists of the selected reviews and included studies. Additionally, we consulted content experts in the field to confirm the final list of included studies. The original search included articles published up until February 2019; an updated search was performed in December 2020. We searched the Embase (1974–present), Medline (1946-present), PsycINFO (1987–present), PubMed (1950–present), CINAHL (1937–present), and SPORTDiscus (1837–present) databases using the following keywords (for a sample search strategy see S2 File): (1) Terms for interventions included: 'intervention stud*' OR 'program' OR 'curriculum' OR 'physical education' OR 'promotion' OR 'initiative' OR 'behaviour change' OR 'strateg*', (2) terms for strength training included: strength training OR resistance training OR muscle strengthening. Grey literature was searched using the Canadian Agency for Drugs and Technologies in Health Grey Matters Tool, the first 10 pages of Google search results, and the Obesity Evidence Hub, Fitness Australia, Physiopedia, and National Academy of Sports Medicine websites. Given the resources available, we limited the search to include only articles written in or translated into English.

## Study selection

Eligible studies: (a) were original peer-reviewed articles or grey literature, (b) used any intervention study design, (c) included behavioural interventions targeted towards improving strength training participation, and (d) measured strength training participation, including adherence and attendance, by direct observation, self-reporting, or objective measures. Articles with a measure of strength training participation as an outcome were included to help distinguish *behavioural* interventions designed to improve strength training participation from interventions designed to improve health outcomes. Non-eligible studies: (a) did not include a strength training–only group, (b) employed interventions targeting multiple health behaviours simultaneously (e.g., diet, self-management, etc.), and (c) assessed health outcomes but did not include strength training behaviour outcomes.

Duplicate articles were removed, and the remaining titles and abstracts were screened for eligibility. Relevant articles had their full texts reviewed for inclusion/exclusion criteria. Reasons for excluding studies are documented in the Open Science Framework repository. The first author (JM) and co-author (STh) screened all the remaining articles independently and resolved discrepancies through discussion. If no consensus was reached, LL acted as third reviewer to resolve discrepancies.

## Data extraction

Researchers and patient/healthcare provider partners jointly developed a data extraction form using Microsoft Excel. To calibrate the data extraction methods, JM and STh independently and iteratively extracted eight articles over three meetings to reach consensus in the data charting approach. STh extracted the remaining articles and JM checked the extraction. The following were extracted from each study: study purpose, study design, country, population, sample size, strength training behaviour measure, theory used, intervention mode of delivery, provider, setting, use of group or individual delivery, intervention duration, exercise frequency, intensity, volume, and intervention procedure. Data were then charted to summarize frequencies of the extracted content.

**Coding for behaviour change techniques (BCTs).** Interventions were coded for BCTs using the 93 Behaviour Change Technique Taxonomy version 1 (BCTTv1) [33]. Authors JM and JL completed a BCTTv1 online training program and developed and piloted a coding manual (see S3 File) prior to coding (www.bct-taxonomy.com). The BCTTv1 has previously demonstrated support for good inter-coder and test-retest reliability [38]. JM and JL independently coded each study and resolved discrepancies through discussion.

## Data analysis and presentation

Quantitative analysis (e.g., frequency analysis) was conducted on study and intervention characteristics and the use of theory. Study prescription parameters were descriptively reported. Behaviour change techniques used in the interventions were mapped on a grid with BCTs on the y-axis and individual studies on the x-axis.

## Stakeholder engagement

We used the Patient Engagement In Research (PEIR) framework to guide the patient/healthcare provider partners' involvement at relevant stages of the review process to improve the usability of recommendations to end-users [39, 40]. Specifically, Arthritis Research Canada's Arthritis Patient Advisory Board ([APAB] a group of advocates who bring lived experience and patient knowledge to research decision making) was consulted to shape the research question. Two patient/clinician partners (AH, KT) contributed to the interpretation of the findings and development of the paper.

## Results

### Study selection

Twenty-seven interventions targeted towards improving strength training participation met the eligibility criteria (Fig 1). Full citations of the included studies and their companion protocols or follow-up papers are included in S4 File.



Records identified through
database searching
(n = 9004)

Additional records identified
through other sources
(n =32)

Records after duplicates removed
(n = 8072)

Records screened for title
and abstract
(n =8072)

Records excluded
for lacking content related
to strength training
interventions (n =7990)

Full-text articles assessed
for eligibility
(n =85)

Full-text articles excluded
(n =53)
No full text available (n=5)
No strength training only
group (n=40)
No measure of strength
training participation or
behaviour (n=8)

Studies included in
systematic review
(n =32)

**Fig 1. PRISMA flow diagram.**

## Study characteristics

Frequency summaries of study characteristics are provided in Table 1. The study designs were primarily randomized controlled trials, and participant sample sizes ranged from 6 [41] to 3500 participants [42], with a total of 5973 participants across the 27 interventions. The studies were conducted in five different countries, primarily the United States. Study participants varied from people with medical conditions/chronic diseases (e.g., osteoarthritis, cancer, spinal cord injury, Type II diabetes, cardiac conditions) or who were overweight/obese to adults and older adults who were healthy or with a functional disability. Only one study examined children (<18 years [43]). No studies examined adults aged <35 years old. In all 27 studies, the number of participants who engaged in strength training fewer than two times/week (current

**Table 1. Study characteristics and use of theory.**

| | Number of studies (n = 27) | % of studies |
|---|---|---|
| Study design | | |
| Randomized controlled trial | 20 | 74 |
| Single arm pre-post | 3 | 11 |
| Quasi-experimental pre-post | 2 | 7 |
| Quasi-experimental prospective follow-up | 1 | 4 |
| 2x2 factorial trial | 1 | 4 |
| Country | | |
| USA | 18 | 67 |
| Canada | 4 | 15 |
| Australia | 2 | 7 |
| Japan | 2 | 7 |
| Belgium | 1 | 4 |
| Population | | |
| Older adults ($\geq$65) | 6 | 22 |
| Children ($\leq$18) | 1 | 4 |
| General | 4 | 15 |
| Knee osteoarthritis | 4 | 15 |
| Breast cancer survivors | 3 | 11 |
| Multiple conditions | 3 | 11 |
| Spinal cord injury | 1 | 4 |
| Cardiac rehabilitation patients | 1 | 4 |
| Overweight and obese | 1 | 4 |
| Cancer survivors | 1 | 4 |
| Type II diabetes | 1 | 4 |
| Prostate cancer | 1 | 4 |
| Sample size | | |
| <20 | 2 | 7 |
| 20–50 | 7 | 26 |
| 51–100 | 7 | 26 |
| >100 | 5 | 19 |
| >200 | 6 | 22 |
| Mean age[*] | | |
| $\leq$18 | 1 | 4 |
| 19–35 | 0 | 0 |
| 36–64 | 13 | 48 |
| $\geq$65 | 12 | 44 |
| [*]one study did not report age | | |
| Theory[$] | | |
| Social cognitive theory | 9 | 33 |
| Transtheoretical model | 4 | 15 |
| Self-determination theory | 2 | 7 |
| Other non-behavioural theory | 1 | 4 |
| None | 12 | 44 |
| Strength training behaviour measure[$] | | |
| Attendance | 13 | 48 |
| Exercise log | 11 | 41 |
| Survey | 9 | 33 |

(*Continued*)

**Table 1.** (Continued)

| | Number of studies (n = 27) | % of studies |
|---|---|---|
| Direct observation | 1 | 4 |
| Timeline follow-back | 1 | 4 |

Note.

*one study did not report age;

$some studies fulfilled multiple criteria for a given category.

Percentages are rounded to the nearest whole number.

strength training guidelines) at baseline was not defined. A total of 14 studies (in addition to one that employed social marketing principles) used behaviour change theory to guide intervention development. Most studies employed social cognitive theory [44–54]. The others used the transtheoretical model [54–57] or self-determination theory [58–60]. All the studies employed some form of self-report measure to assess strength training behaviour.

## Intervention characteristics

Most interventions were delivered via supervised exercise sessions, education sessions, and individual counselling. A variety of individual, group, and mixed (group and individual) delivery settings were employed. Almost all interventions used face-to-face delivery with the exception of Falcon et al. (2014), who used a DVD, and Mailey et al. (2020), who used email and print material. Exercise specialists (under a variety of titles such as personal trainer, fitness instructor, exercise instructor, exercise physiologist, physiotherapist, and kinesiologist) were the most common interventionists. Of the 23 interventions delivered by an exercise specialist, only seven reported the qualification of those professionals. The majority of interventions were delivered in community or home settings. The duration of interventions (classed as any form of contact between participant and intervention deliverer) varied from a single contact to two years, with the most common contact frequency and session duration being 1–2x/week for approximately one hour. For a summary of intervention characteristics, see Table 2. For intervention descriptions (i.e., any contacts made between interventionists and participants), see Table 3.

## Prescription parameters

Prescribed exercise frequencies varied from 1x to 4x/week, with most being 2–3x/week. Volume per session ranged from 1 to 3 sets, at 8–15 reps, for 6–12 exercises, with a variety of intensity metrics prescribed, including the Borg 20- and 10-point Rating of Perceived Exertion, percentage of one repetition maximum, time under tension, self-created intensity metrics, and completion of exercise to momentary muscle failure (Table 3).

## Behaviour change techniques

Inter-coder agreement for the BCT coding was 90% (Kappa = 0.95, prevalence-adjusted and bias-adjusted kappa [PABAK] = 0.97). Kappa values greater than 0.81 are considered 'almost perfect' strength of agreement [61]. Studies that employed BCTs in both the intervention and control condition are presented in the Open Science Framework repository without BCTs that were common among both groups.

Of the potential 93 BCTs, 39 were included across the studies. A range of two to 19 BCTs were used within individual studies. The most common BCTs (i.e., those used in at least half of the studies) included instructions on how to perform a behaviour, a credible source, adding objects

**Table 2. Intervention characteristics.**

|  | Number of studies (n = 27) | % of studies |
|---|---|---|
| Mode of delivery* |  |  |
| Face-to-face | 25 | 93 |
| Print material | 11 | 41 |
| Telephone | 9 | 33 |
| DVD/video | 5 | 19 |
| Email | 2 | 7 |
| Website | 2 | 7 |
| Video conference | 1 | 4 |
| Provider* |  |  |
| Personal trainer/fitness specialist/exercise instructor | 17 | 63 |
| Researcher | 5 | 19 |
| Physiotherapist | 2 | 7 |
| Physical education teacher | 2 | 7 |
| Exercise physiologist | 2 | 7 |
| Community leader | 1 | 4 |
| Health worker | 1 | 4 |
| Professional | 1 | 4 |
| Peer | 1 | 4 |
| Health educator | 1 | 4 |
| Kinesiologist | 1 | 4 |
| Setting* |  |  |
| Home | 13 | 48 |
| Community fitness centre | 6 | 22 |
| University | 5 | 19 |
| Seniors' centre/retirement home | 4 | 15 |
| City-wide | 1 | 4 |
| School | 1 | 4 |
| Group/individual |  |  |
| Individual | 11 | 41 |
| Group/individual | 8 | 30 |
| Group | 7 | 26 |
| Community-wide | 1 | 4 |
| Intervention duration |  |  |
| <6 months | 15 | 56 |
| 6–12 months | 9 | 33 |
| >1 year | 3 | 11 |
| Intervention procedure* |  |  |
| Supervised sessions | 21 | 78 |
| Telephone calls/counselling | 9 | 33 |
| Education sessions | 8 | 30 |
| Information resources | 5 | 19 |
| Home visits | 4 | 15 |
| Mass/individual encouragement activities | 4 | 15 |
| Watching DVDs | 1 | 4 |

Note.

*Some studies fulfilled multiple criteria for a given category.

**Table 3. Recommended exercise prescription parameters and intervention descriptions.**

| Author (year) | Recommended exercise prescription parameters | | | Intervention descriptions |
|---|---|---|---|---|
| | Exercise frequency | Exercise volume | Exercise intensity/progression | |
| Baker et al. (2001) | 3x/week | 2 sets x 12 reps x 7 exercises | 3–5 on 10-point Borg scale progressed to 8 | Home visits |
| | | | | Weeks 1–3: 2x/week |
| | | | | Week 4: 1x/week |
| | | | | Weeks 5–16: 1x/2weeks |
| Baker et al. (2020) | 2x/week | 2 sets x 8–15 reps | "Somewhat hard level of intensity" | Weeks 0–6 (run-in period): Group exercise |
| | | | | Months 0–6: Weekly telephone counselling |
| | | | | Months 7–24: Monthly telephone counselling |
| Falcon (2014) | 2x/week | 30 minutes | Gradual progression (no specifics provided) | Watching DVD 2x/week @ 30 minutes |
| | | | | Weekly telephone calls |
| Fetherman, Hakim & Sanko (2011) | 3x/week | Up to 3 sets x 10 reps | Easy (2) progressed to moderate difficulty (4) on a 5-point strength intensity scale | Intro education session @ 1 hour |
| | | 60 minutes | | Individual counselling session @ 10 minutes, 2 days/week |
| | | | | On-site supervised exercise session @ 1 hour, 2 days/week |
| Jette et al. (1998) and Jette et al. (1999) | 3x/week | 35 minutes | Instructed to increase resistance when they could perform 10 repetitions of a movement pattern without significant fatigue or loss of proper execution | 2 home visits |
| | | | | 7 or 8 telephone contacts |
| Kamada (2013) & Kamada et al. (2015) | Not reported | Not reported | Not reported | Leaflets and flyers distributed to 4036 households at least twice |
| | | | | Posters hung at 276 sites |
| | | | | Banners placed in all community centres |
| | | | | Audio messages (60–90 seconds long) broadcasted to each household 12 times |
| | | | | Mass and individual encouragement activities conducted by professionals 142 times |
| | | | | Call centre |
| Latimer-Cheung et al. (2013) | 3x/week | ~30 minutes | Not reported | Single visit @ 70 minutes +/-19.52 |
| Lubans, Mundey, Lubans & Lonsdale (2013) | 2x/week | 2 sets x 10–15 reps x 10 exercises | 12–16 on the 20-point Borg scale | Information session @ 10–15 minutes 2x/week |
| | | 45–60 minutes | | Supervised sessions @ 45–60 minutes 2x/week |
| Lubans, Plotnikoff, Jung, Eves & Sigal (2012) and Plotnikoff et al. (2010) | 3x/week | Week 1: 2 sets x 10–12 reps | Week 1: 50–60% 1RM | Supervised home sessions |
| | | | Week 5: 70–80% 1RM | Weeks 0–2: 3x/week |
| | | | Week 9: 70% 1 RM | Weeks 3–4: 2x/week |
| | | Week 2: 3 sets x 10–12 reps | Week 10: 70–85%1RM | Weeks 5–8: 1x/week |
| | | Week 9: 2 sets x 8–10 reps | Week 16: 80%1RM | Week 9–16: 1x/2weeks |
| | | Week 10: 3 sets x 8–10 reps | | |
| | | Week 16: 2 sets x 8–10 reps | | |
| Mailey et al., 2020 | NR | 2 sets x 8–15 reps | NR | Receipt of a strength training workout plus educational materials |

(*Continued*)

**Table 3.** (Continued)

| Author (year) | Recommended exercise prescription parameters | | | Intervention descriptions |
|---|---|---|---|---|
| | Exercise frequency | Exercise volume | Exercise intensity/progression | |
| Mikesky et al. (2006) | 3x/week | 3 sets x 8–10 reps | Maximum resistance that could be lifted within prescribed reps Progression to greater resistance levels was implemented when the participant could perform 12 repetitions on the last training set for 2 consecutive workouts | Supervised training sessions (1 hour) |
| | | | | Months 0–3: 2x/week |
| | | | | Months 4–6: 1x/week |
| | | | | Months 7–9: 2x/month |
| | | | | Months 10–12: 1x/month |
| | | | | Contacts from fitness trainer after missed sessions, newsletter, buddy system, group training sessions, social gatherings (frequency NR) |
| Mikesky, Topp, Wigglesworth, Harsha, Edwards (1994) | 3x/week | Week 1: 1 set | Participants were instructed to move to the next larger tubing size when they could perform 12 repetitions with good exercise form during their last set. | Supervised exercise classes @ 1x/week |
| | | Week 3: 3 sets of lower body and 2 sets of upper body exercises | | |
| | | 55 minutes | | |
| Millen & Bray (2009) | 2x/week progressing to 3x/week | 3 sets x 10–15 reps x 6 exercises | Steady progression | Orientation session |
| | | | | Education and supervised exercise @2–3x/week |
| Mullane, Bocchicchio & Crespo (2017) | 2x/week | 45 minutes, supervised | 60 seconds time under tension, increasing by 10 seconds every 4 sessions up to 90 sessions | Supervised sessions plus educational quizzes and games 2x/week @ 45 minutes |
| Osuka et al. (2017) | 1x/week | 15–20 reps x 6 exercises | Borg rating of perceived exertion (6–20) of 13 "somewhat hard" or higher | Supervised exercise session 1x/week @70–100 minutes |
| | | 50–100 minutes | | |
| Ott et al. (2004) | 2x/week | 2 sets x 8 reps x 9 exercises | Weight progression over the 6 months was individualized based on size, age, and strength at initiation of the study | Home visits/phone calls at baseline x 2 + monthly |
| | | 50 minutes | | |
| Papadopoulos & Jager (2016) | 2x/week | 1 hour | Started with least resistive tube during initial meeting, if no soreness was experienced, advised to progress to the next level | Supervised sessions @ 2x/week |
| | | | Larger tubing sizes were used once participants could perform 12 repetitions with proper exercise form during their last set | Education programs @ 1x/week |
| Schmitz et al. (2007) and Arikawa, O'Dougherty & Schmitz (2011) | 2x/week | Months 1–4: 60–90 minutes | Gradual progression with highest weight lifted for 2 sets maintained | Weeks 0–16: supervised sessions @ 2x/week |
| | | Months 5–24: 45 minutes | | Week 17–year 2: booster sessions every 12 weeks |
| | | | | Fitness trainers available for contact, study website (frequency NR) |
| | | | | Social gatherings @ 2x/year |
| | | | | Newsletter @ 1x/month |
| Schwartz & Winters-Stone (2009) | 4x/week | Variable over the course of the study between 3 sets of 12 reps to 2 sets of 18–20 reps x 6–8 exercises | Undefined %1-RM | Telephone calls |
| | | 20–30 minutes | | Month 1: 1x/week |
| | | | | Months 2–3: 1x/2weeks |
| | | | | Months 4–12: 1x/month |

*(Continued)*

**Table 3.** (Continued)

| Author (year) | Recommended exercise prescription parameters | | | Intervention descriptions |
|---|---|---|---|---|
| | Exercise frequency | Exercise volume | Exercise intensity/progression | |
| Sigal et al. (2007) | 3x/week | 2–3 sets x 7–9 reps x 7 exercises | Weight was increased by 5–10 pounds when the participant could perform more than 8 repetitions of a given exercise while maintaining proper form, and vice versa | Individual meetings @ 15–45 minutes |
| | | 15–45 minutes | | Month 1: 1x/week |
| | | | | Months 2–3: 1x/2weeks |
| | | | | Months 4–6: 1x/month |
| Sparrow, Gottlieb, DeMolles & Fielding (2011) | 3x/week | 2 sets x 12 reps x 8 exercises | Started with lowest resistance, increased by 2 pounds each succeeding session, provided the participant was able to complete 2 sets of 10 or more repetitions. | Supervised sessions 1x/week @ 1 hour |
| | | 60 minutes | | |
| Teychenne et al. (2015) | 3x/week | 45–60 minutes | Continual progressive overload (increments of 2–10%) | Supervised sessions 2x/week @ 45–60 minutes |
| | | | | Instructional newsletters: months 2, 4 and 6 |
| | | | | Motivational incentives: months 0 and 2 |
| | | | | Behavioural telephone counselling: progressed from weekly, to bi-weekly, to monthly to bi-monthly for 6-months |
| Vanroy et al. (2019) | 4x/week | 20 minutes, 8 exercises | NR | 4 exercise sessions/week |
| | | Week 1: 8 reps | | |
| | | Week 2: 10 reps | | |
| | | Week 3: 2 sets x 8 reps | | |
| Wilson, Strayer, Davis & Harden (2018) | 2x/week | 8 exercises | NR | Supervised exercise sessions @ 1 hour 2x/week |
| | | 60 minutes | | |
| Winett et al. (2015); Williams et al. (2016); Davy et al. (2016) and Marinik, Kelleher, Savla, Winett & Davy (2014) | 2x/week | 8–12 reps x 12 exercises | Moderate effort to concentric failure | Orientation sessions x 2 |
| | | 35–45 minutes | | Supervised exercise sessions 2x/week @ 35–45 minutes for 3 months |
| | | | | Orientation sessions in new facility x 3 |
| | | | | Ongoing continuous online feedback |
| Winters-Stone et al. (2011) Winters-Stone et al. (2012) | 3x/week | 1–3 sets x 8–12 reps x 7–9 exercises | 60–70% of 1-RM | Supervised exercise sessions @45–60 minutes, 2x/week |
| | | 45–60 minutes | | |
| Winters-Stone et al. (2016) | 2x/week | 8–15 reps x 8–10 exercises | 4–15% of body weight (lower body exercises) and a weight that could be lifted for 15 reps progressed to 8 reps (upper body) | Supervised exercise sessions 2x/week @ 1 hour |
| | | 1 hour | | |

Note: FU = follow-up; reps = repetitions, RM = repetition maximum, NR = not reported.

to the environment, behavioural practice/rehearsal, graded tasks, behavioural goal setting, feedback on behaviour, self-monitoring of behaviour, and practical social support (Fig 2). Coders of this review also explored common themes outside of the BCT taxonomy and found consistent mention of tailoring, fun, and variety as strategies to support the goal of the intervention.

## Discussion

The review identified 27 unique interventions for improving strength training participation conducted to date. Social cognitive theory, the transtheoretical model, and self-determination

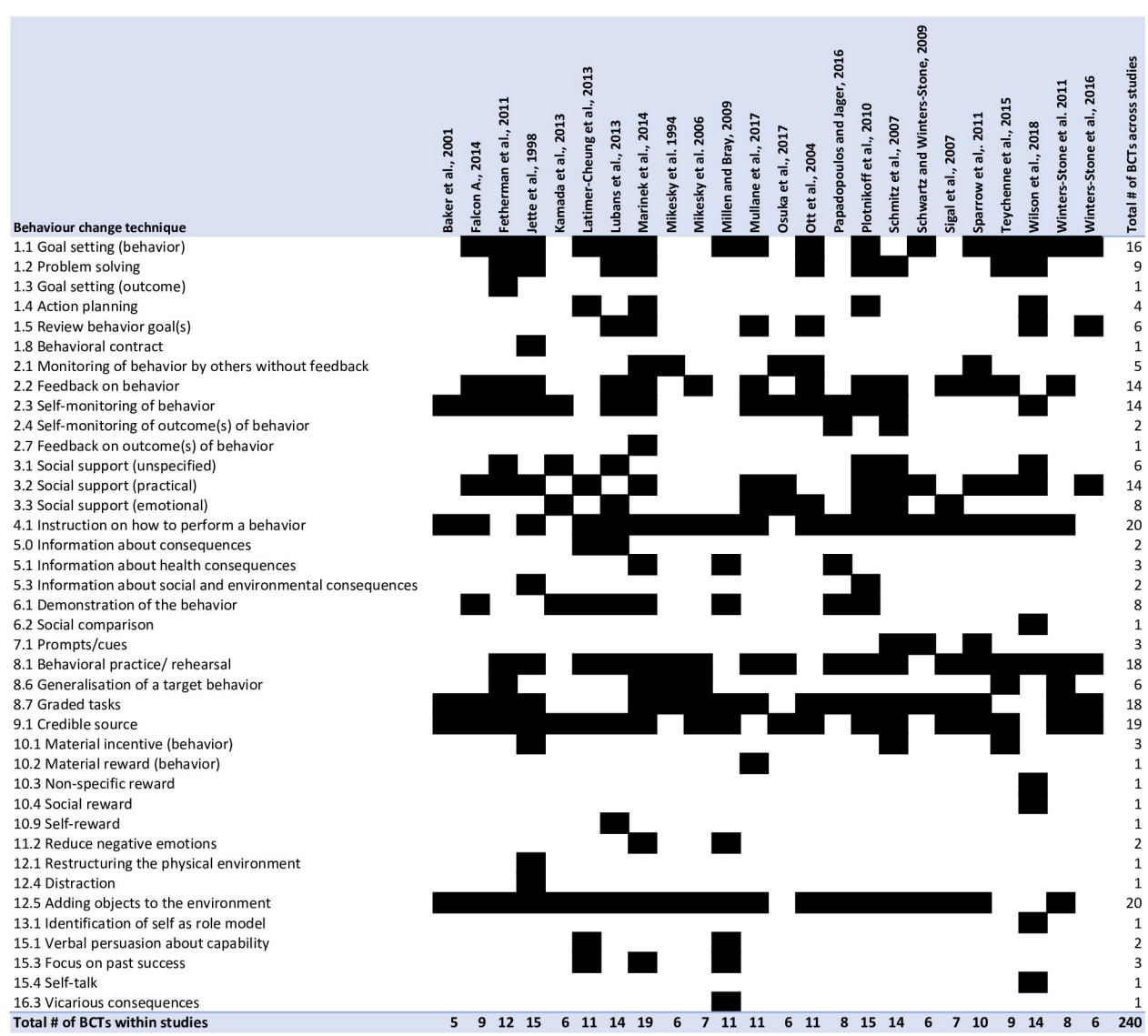

**Fig 2. Summary of BCTs.** Note: A shaded cell indicates a given BCT was present.

theory were the only behaviour change theories employed across the studies. Almost all interventions were delivered face-to-face and by an exercise specialist in community or home settings. Instructions on how to perform the behaviour, adding objects to the environment, a credible source, behavioural practice, graded tasks, and goal setting were the most commonly used BCTs and may be core components of interventions to improve strength training participation. The extent to which BCTs and intervention characteristics (or the synergy between them) influence the effectiveness of interventions merits further study. It should also be noted that many other BCTs, intervention characteristics, and theories have yet to be studied.

## Use of theory

Social cognitive theory, self-determination theory, and the transtheoretical model were the only behavioural theories applied in the studies in this review. It has been previously supported

that theory-based behavioural interventions are more effective than atheoretical interventions [30, 32, 62]. Given that there are over 83 theories of behaviour change, other theories may be suitable, or more suitable, for designing interventions to improve strength training participation [63]. Indeed, within the general physical activity literature, successfully tested theories span the social cognitive, humanistic, dual process, and socioecological theoretical frameworks (for a review, definitions, and example theories within these frameworks see [64]). To date, the theories tested in changing strength training behaviour fall within the social cognitive (social cognitive theory, transtheoretical model) and humanistic (self-determination theory) frameworks. These theories align with some of the factors previously identified as effective in influencing strength training participation, including self-efficacy (confidence to perform strength training), subjective norms (belief that others support the behaviour), affective judgments (positive or negative feeling states as a result of the behaviour), and intentions [25]. However, they fall short of bridging the intention-behaviour gap (e.g., targeting self-regulation), explaining non-conscious or automatic processes (dual process frameworks), or exploring the role of environmental factors (socioecological frameworks) [25, 64]. While it is too early to make explicit recommendations about which theories to test in the future, it is important to stress at this point that participating in physical activity, including strength training, is complex and likely requires explanation that spans across the socialcognitive, humanistic, dual process, and socioecological theoretical frameworks [64].

### Intervention characteristics

**Exercise prescription.** Prescriptions for sets, reps, intensity, and/or progression were highly variable across the studies. No interventions examined the impact of prescription parameters on strength training participation. The perception that strength training prescriptions are complex and must include the use of heavy weights is an acknowledged barrier to participation in strength training [25, 27, 28, 65]. More research is needed to understand how to make strength training prescriptions more accessible (i.e., simple to understand and easy to do). For example, the American College of Sports Medicine in conjunction with the American Heart Association have suggested a *minimum* prescription of 1 set of 8–12 reps to volitional fatigue of 8–10 resistance exercises that target the major muscle groups [66]. A 10-week strength training program employing these recommendations at varying frequencies among 1619 adults and older adults demonstrated a 95% satisfaction rate with a 91% completion rate [67]. More simplified strength training regimes involving single sets at high intensities, completed in as little as 20 minutes, have also been found to offer benefits that may differ only marginally from the benefits derived from complex weightlifting protocols at the public health level [29, 68, 69]. Furthermore, recent work has challenged mainstream prescription principles such as rep ranges that target strength, hypertrophy, and muscular endurance and suggests that these outcomes may be obtained, and even optimized, across a spectrum of rep ranges [70]. Likewise, performing strength training to failure vs. non-failure has been shown to produce similar strength and hypertrophy adaptations [71]. These recent findings provide further support for the use of less complicated or intimidating strength training prescriptions. Reducing the time commitment and complexity of strength training prescriptions may play a critical role in changing population-level strength training behaviour. Overall, however, strength training prescriptions and their influence on strength training participation are poorly understood.

**Mode of delivery.** With the exception of two studies, all interventions were delivered in-person. Specifically, Falcon et al. (2014) used a DVD and weekly telephone calls to deliver a 12-week home-based strength training program and Mailey et al. (2020) used email and print material. Some studies incorporated the use of telephones, video conferencing, print resources,

email, websites, and DVDs, but all had face-to-face contact at one or more points in the intervention. This approach differs from that of general physical activity programs, where an increasing number of interventions are delivered remotely via websites, telephones, or mobile apps [62, 72, 73]. A possible reason for the ubiquity of face-to-face strength training interventions is that trainers need to demonstrate and provide feedback on technique, which they do not need to do for simple aerobic activities such as brisk walking. More research is needed to understand whether in-person interventions specific to strength training are necessary or whether technology (e.g., online videos, video conferencing, mobile apps) can replace person-to-person contact. Remote delivery of strength training may be a particularly timely topic of research given that current COVID-19 disease control measures can also improve the accessibility of health service delivery moving forward.

**Interventionists.** All but four studies employed exercise specialists as interventionists. No studies compared the effectiveness of exercise specialists to the effectiveness of other healthcare providers or peers. Nurse- and physician-led interventions have been shown to be both cost-effective and successful in improving general physical activity participation [74, 75]. Likewise, peers have contributed to general physical activity improvements and in some cases have been identified as preferred messengers [76, 77]. It is possible that healthcare providers and peers, or others who are not professionally trained as exercise specialists, may be suitable choices for delivering generalized strength training interventions. However, it is likely that non-exercise specialists would require additional training to ensure the safety of clients engaging in strength training participation [78], particularly those with specific health conditions. The need and potential for non-exercise specialists to improve strength training participation remains unstudied.

**Other intervention characteristics that influence affect.** Affect, described as a positive or negative arousal state or dimensions of pleasure and displeasure, may be another important intervention target for strength training participation interventions. Common themes coded outside of the BCT taxonomy included fun and variety as goals of the intervention. Strength training has been described as "boring" compared to aerobic exercise, while the inclusion of novelty and variety has been shown to influence motivation and participation in physical activity [79, 80]. Future research could examine whether adding variety to strength training exercise prescriptions (e.g., using a variety of exercises, changing exercise order, number of repetitions, etc. [28]) promotes more positive affect and subsequently participation. Performing strength training in a group setting or considering alternative muscle strengthening activities may also positively influence affect and, ultimately, strength training participation. Specifically, strength training participation may be higher in a group setting vs. an individual setting, as demonstrated by Fetherman et al. (2011), who showed 88% adherence to strength training in a group setting compared to 49% in an individual setting over the same period [55]. Sports are also often overlooked as forms of muscle strengthening activities. In a Scottish national survey of strength training participation, certain sports—including athletics, canoeing/kayaking, climbing, horse riding, rowing, skiing/snowboarding, swimming, and waterskiing—were considered muscle strengthening activities [15]. The influence of affect has been supported by a systematic review of factors associated with strength training behaviour [25]. It should be noted that although affect may play an important role in strength training participation, strength training intensity should still be promoted at an adequate level to achieve health benefits [27].

## Behaviour change techniques

The majority of the most commonly employed BCTs were similar to those found to be most effective in the general physical activity literature (i.e., goal setting [behaviour], feedback on

behaviour, self-monitoring of behaviour, social support [practical], instructions on how to perform a behaviour, behavioural practice/rehearsal, graded tasks) [31, 32, 81]. Adding objects to the environment (e.g., provision of exercise equipment) and using a credible source (e.g., use of exercise specialists to deliver the intervention) were other commonly used BCTs in strength training participation interventions. These BCTs differ from the most commonly employed BCTs in the general physical activity literature and may be unique to strength training (likely for the reasons related to complexity and exercise specialist delivery summarized above). Despite the similarity of BCTs in the strength training and general physical activity literature, we cannot infer that strength training interventions are the same as general physical activity interventions. BCTs describe the individual components that comprise an intervention but do not include details on how those BCTs are implemented (e.g., the dose, frequency, and mode of delivery). For example, because of the complexity of strength training, prolonged and more comprehensive feedback on behaviour and behavioural practice may be required compared to aerobic exercise, or instructions on how to perform the behaviour may require a mode of delivery with a visual component.

The BCTs listed in this scoping review are the most commonly used ones; however, effectiveness cannot be inferred from frequency [30]. It is possible that other less frequently studied, or even untested, BCTs are effective for changing strength training behaviour. Future research should explicitly examine the interaction between BCTs, how they are delivered, and the effects on strength training participation.

## Strengths/limitations

A strength of this scoping review was the engagement of end-users. Patient/healthcare provider perspectives were integrated into the design, interpretation, and draft and revision of this review. This integrated knowledge translation approach helped to shed light on accessible language, clinical and patient perspectives, and linked findings grounded in theory to applications relevant in the real world. Furthermore, by using the Behaviour Change Technique Taxonomy V1 to code interventions, we made our findings accessible to researchers in a variety of disciplines, who may use them to guide future research in this nascent field.

A few limitations must be acknowledged. First, we could have included a much larger number of studies that combined a balance training component with strength training. However, the focused inclusion/exclusion criteria were developed to address an understudied area and understand the intervention components that are unique to strength training. In addition, alternative forms of exercise that may qualify as muscle strengthening (e.g., yoga, calisthenics, and Pilates) were not included in the search and may be important to examine in the future. Second, this review was originally registered with PROSPERO (CRD42019120251) as a systematic review with two research questions. Question one ("what are the effects of strength training participation interventions?") will be addressed in a separate systematic review. Question two in the registered protocol was defined as "what BCTs, theories, and modes of delivery are used in strength training interventions currently?'. Given the nascency and heterogeneity of the literature, we changed to a scoping review methodology for this specific question. Specifically, it was more appropriate to map the current state of the literature and suggest a research agenda that addresses current gaps than to assess the effectiveness of these intervention components, therefore warranting a scoping review methodology.

## Conclusion

This review highlights several understudied intervention components that have the potential to considerably impact strength training behaviour change and merit exploration. Potential

topics for future exploration include i) exploring theory that extends beyond the social cognitive and humanistic frameworks to include dual process or socioecological frameworks, ii) how prescription parameters can be modified to promote increased participation without sacrificing effectiveness, iii) whether these interventions can be delivered by non-exercise specialists such as clinicians and peers or by using remote delivery, iv) how interventions can target positive affect to influence strength training participation, and v) how to optimize the selection and dosing of BCTs. Separating strength training from aerobic interventions acknowledges the barriers and strategies that are unique to strength training participation. With an increased research focus on strength training behaviour change specifically, population participation in meeting *both* strength and aerobic exercise guidelines to optimize population health outcomes may be improved.

## Supporting information

**S1 File. PRISMA-ScR checklist.**
(DOCX)

**S2 File. Search strategy.**
(DOCX)

**S3 File. Behaviour change technique coding manual.**
(PDF)

**S4 File. List of included studies.**
(DOCX)

## Acknowledgments

We would like to thank Dr. Richard Winett for his review of the included studies and Arthritis Research Canada's Arthritis Patient Advisory Board for their contributions in developing the research question and identifying future directions for research.

## Author Contributions

**Conceptualization:** Jasmin K. Ma, Stephanie Therrien, Alison M. Hoens, Karen Tsui, Linda C. Li.

**Data curation:** Jasmin K. Ma, Jennifer Leese, Stephanie Therrien, Alison M. Hoens, Karen Tsui, Linda C. Li.

**Formal analysis:** Jasmin K. Ma, Jennifer Leese, Stephanie Therrien, Alison M. Hoens, Karen Tsui, Linda C. Li.

**Funding acquisition:** Jasmin K. Ma, Alison M. Hoens, Karen Tsui, Linda C. Li.

**Investigation:** Jasmin K. Ma.

**Methodology:** Jasmin K. Ma, Alison M. Hoens, Karen Tsui, Linda C. Li.

**Project administration:** Jasmin K. Ma, Linda C. Li.

**Resources:** Jasmin K. Ma, Linda C. Li.

**Supervision:** Jasmin K. Ma, Linda C. Li.

**Writing – original draft:** Jasmin K. Ma, Jennifer Leese.

**Writing – review & editing:** Jasmin K. Ma, Jennifer Leese, Stephanie Therrien, Alison M. Hoens, Karen Tsui, Linda C. Li.

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
