## [Decision Letter · Decision Letter 0]

29 Oct 2020

PONE-D-20-27748

A scoping review of strength training behaviour change interventions: Future research and practical applications

PLOS ONE

Dear Dr. Ma,

Thank you for submitting your manuscript to PLOS ONE. After careful consideration, we feel that it has merit but does not fully meet PLOS ONE’s publication criteria as it currently stands. Therefore, we invite you to submit a revised version of the manuscript that addresses the points raised during the review process.

Understanding the use of behaviour change theory in strength training is an important and understudied topic. The paper has been assessed by two experts in the field of strength training and behaviour change theory and they have raised some excellent points. While the paper identifies an important area of research, the paper requires much editing before it can be considered eligible for publication. Below you will find comments from the reviewers, as well as some additional points of my own. If you feel that you can make the required changes, we would be happy to re-review the paper.

We look forward to receiving your revised manuscript.

Kind regards,

Stephanie Prince Ware, PhD

Academic Editor

PLOS ONE

Journal Requirements:

2. We noted that your search was completed in February 2019. Please ensure that your search is updated and any relevant articles published since February 2019 are included in your review.

Additional Editor Comments (if provided):

1. Overall, consider removing the “ST” acronym for strength training as it is not commonly used in the field.

2. Please use acronyms consistently throughout the text. For example, line 395 BCT is written out in full in the same paragraph where the acronym appears and the PA acronym appears on line 400 without previously being introduced.

3. For numbers under 10 that appear in text (i.e. not in tables) please write out full. For example, line 249 please change “2” to “two”.

4. Were only 24 strength training interventions described in the literature or only 24 that described using a theory of behaviour change?

5. The search is now almost two years old, have the authors considered an update?

6. Why were studies targeting multiple health behaviours simultaneously excluded rather than a sensitivity analysis undertaken?

7. The description for the grey literature search methods is missing.

8. The use of the BCTTv1 training was a nice addition to the methodology.

9. Line 265, please place the examples in parentheses.

Reviewers' comments:

Reviewer's Responses to Questions

**Comments to the Author**

1. Is the manuscript technically sound, and do the data support the conclusions?

Reviewer #1: Partly

Reviewer #2: Partly

2. Has the statistical analysis been performed appropriately and rigorously? 

Reviewer #1: N/A

Reviewer #2: Yes

3. Have the authors made all data underlying the findings in their manuscript fully available?

Reviewer #1: Yes

Reviewer #2: Yes

4. Is the manuscript presented in an intelligible fashion and written in standard English?

Reviewer #1: No

Reviewer #2: Yes

5. Review Comments to the Author

Reviewer #1: Review comments to authors:

The current study is on a topic of relevance and the idea is innovative. This scoping review has salience towards the planning of behaviour change interventions for strength training.

Of greater importance will be the systematic review regarding the efficacy of these interventions which is currently being under-taken (as indicated by the authors in the Discussion section). One option is to combine both papers or submit as companion papers. There is probably enough information focus to warrant both papers.

The methodological approach for this scoping review appears to have some limitations. It is not clear how the authors selected the search terms; more detail is warranted. Why were other relevant databases not selected (e.g. SPORTDiscus, Cinahl, Scopus)? We believe that some key studies have not been included in this review. It was not entirely clear how the authors selected the included studies, i.e. clarification is required regarding the selection criteria for study selection. On page 8 – it is reported that ‘any study design; was included in the search strategy. However this paper reports on ‘intervention studies’.

Overall, the manuscript requires major revisions including more focused writing and structuring within each section. Specific examples and suggestions are detailed below.

Introduction

Overall, the introduction needs to be written in more focused and clear language. For example, the first sentence should be edited to “International guidelines recommend that adults and older adults should engage in muscle strengthening activities at least twice weekly.” The sentence starting on line 98 also needs editing, it should read “ST is especially important for older adults, where increased strength helps support continued function and independence”.

Instead of using the term “ST behaviour change interventions”, which reads quite awkwardly. It is recommended that authors reword this to “ST interventions that use/include/utilizes behaviour change strategies”.

It is also recommended that when authors discuss the low rates of meeting the ST guidelines, that they compare these to the prevalence of meeting the aerobic guidelines.

On a number of locations, the authors mention ‘strengthening activities’, e.g. line 106. It is recommended that the authors reword this to ‘muscle strengthening’ or ‘muscle strengthening activities’ and use this throughout the manuscript.

The sentence starting on line 110 is awkward and requires rewording. For example, “When compared to aerobic activities (e.g. walking), the perceived knowledge and effort required to effectively perform muscle strengthening activities (e.g. specific techniques, cost of equipment, knowledge of how much ST to perform/ how to progress) can be deterring.” Authors also need to reference this sentence.

The sentence starts on line 113 is awkward and needs rewording. The authors need to be really clear what they are referring to i.e. interventions targeting a particular type of exercise. For example, “Given the unique barriers to ST, it may be that interventions targeting ST require different behaviour change strategies than those targeting aerobic exercise or ST combined with other physical activities.”

The last paragraph of the introduction is awkward and repetitive in places, please revise.

Methods

The method needs to be written in more focused language and be more structured. For example:

It is recommended that authors use an appropriate subheading for the first paragraph.

Page 6. This paragraph is confusing and wordy, please revise using more focused language.

Page 6. The last sentence of the paragraph should be removed.

It is also recommended that authors remove the ‘Stage 1’, ‘Stage 2’ etc from each of the subheadings as it is expected that each section follows in a chronological fashion.

Page 8. In the study selection. While the authors include the inclusion criteria and exclusion criteria. Authors also state on line 191 “Several strength training interventions have been conducted to date, the inclusion of articles with a measure of ST participation as an outcome was selected to help distinguish behavioural interventions from general ST training interventions”. Please clarify this.

On page 9. Authors are using the heading “Charting the data”, please change to heading “Data extraction” as this terminology is used in the abstract. It is important to be consistent with the language used throughout the manuscript.

On page 9, line 215, the authors have written 2.4.1 – please remove.

On page 9, change the subheading “Collating, summarizing, and reporting the results” to “Analysis”

Results

On page 10, under “Study characteristics” and “Intervention characteristics”, instead of spelling out all the information that is already presented in Table 1, do a brief summary. Alternatively, remove Table 1 and Table 2 and keep the text.

Please include the range of participants i.e. ranging between X to 3500.

It is also recommended that the authors edit Table 3. In the first column, add in the reference number as opposed to stating the author and year. In Table 3, the authors mention 1-RM in the ‘Exercise volume’ column and the number of sets in the exercise intensity? 1-RM is a measure of intensity, please go through the table and amend accordingly. In addition, second column under ‘Exercise frequency’, this information is often repeated under the ‘Intervention procedure’ column. Please revise and avoid repeating information.

Discussion

Overall, the discussion requires major revisions, including being written in a more structured and focused language. Some examples listed below:

Page 20, line 302, avoid ‘I’ or ‘We’ statements, please amend to “The present review identified 24 unique interventions for improving ST behaviour”.

In the second paragraph “Opportunity to test theory” the authors are suggesting a number of theories to include, HAPA, SDT and TPB, however the authors do not justify why these ones should be tested? Have they been used in other aerobic and/or aerobic + RT interventions? The authors should remove the sentence on line 321-324 or reword as it doesn’t really fit into the context. All those constructs are part of SCG.

The ‘Intervention characteristics: consideration of practical interventions” paragraph is confusing and hard to follow. The authors needs to rethink their discussion points.

Throughout the discussion, the authors could remove the first sentence under each subheading or remove the subheadings, as the first sentence is repetition from the previous paragraph.

Page 24, line 407, please remove the statement “frequency does not infer effectiveness” unless the authors can reference this. Also reword the next sentence to make it more generalised i.e. other BCT’s may be effective (equally, less or more), but we don’t know as these haven’t been tested.

References

Some references are listed two or more times eg Lubans ,et al 2013. Please go through the reference list and remove duplicates where applicable.

Reviewer #2: A scoping review of strength training behaviour change interventions: Future research and practical applications

REVIEWER COMMENTS

Overview

Thank you for the opportunity to review this manuscript. This paper addresses an important and currently understudied area in physical activity and public health – the use of behaviour change science in strength training. Overall, the paper is generally well-written and provides some interesting and noteworthy findings on an understudied topic.

However, I do have a major concern regarding the search strategy used. For example, reading this, I wondered if the use of only ‘strength training’ OR ‘resistance training’ has resulted in missing some interventions in this area. It should be noted that way strength training is currently defined in the literature is not consistent and may incorporate many different terms: for example: " weight training" "weight lifting" "muscle strengthening" "muscle toning" "weight-bearing training" "weight-bearing strengthening". On this, in my comments below, I have asked the authors to defend the use of this ‘narrow’ search strategy. If they can not, I suggest re-running the search to include a broader search of possible strength training -related terms.

I will now provide my comments on each section of the manuscript in order of presentation.

INTRODUCTION

First, I suggest not abbreviating strength training to ‘ST’ throughout the manuscript. I feel that this would enhance the readability of the manuscript.

Second, this section could be strengthened with the inclusion of more current/relevant references, as well as some edits to improve readability. A recent review article that provides an overview of this strength training in epidemiology/public health and may be useful to read and incorporate into your introduction: https://doi.org/10.1186/s40798-020-00271-w

Specific suggestions

Line 93: ‘International guidelines’ is very vague – I suggest that you change this to ‘International physical activity guidelines for public health’

Lines 96-98 – many of these references are either not current or not that relevant. I suggest including the following:

Cardiometabolic health:

• Ashton RE, Tew GA, Aning JJ, Gilbert SE, Lewis L, Saxton JM. Effects of short-term, medium-term and long-term resistance exercise training on cardiometabolic health outcomes in adults: systematic review with meta-analysis. Br J Sports Med. 2018:bjsports-2017-098970.

• Lemes ÍR, Ferreira PH, Linares SN, Machado AF, Pastre CM, Netto J. Resistance training reduces systolic blood pressure in metabolic syndrome: a systematic review and meta-analysis of randomised controlled trials. Br J Sports Med. 2016:bjsports-2015-094715

TYPE II DIABETES

• Grontved A, Pan A, Mekary RA, Stampfer M, Willett WC, Manson JE, et al. Muscle-strengthening and conditioning activities and risk of type 2 diabetes: a prospective study in two cohorts of US women. PLoS Med. 2014;11(1):e1001587.

• Grontved A, Rimm EB, Willett WC, Andersen LB, Hu FB. A prospective study of weight training and risk of type 2 diabetes mellitus in men. Arch Intern Med. 2012;172(17):1306–12.

ALL-CAUSE MORTALITY

• Saeidifard F, Medina-Inojosa JR, West CP, Olson TP, Somers VK, Bonikowske AR, et al. The association of resistance training with mortality: a systematic review and meta-analysis. Eur J Prev Cardiol. 2019;2047487319850718.

You should also cite some of the recent reviews on mental health and strength training, as these are very important as well

• Gordon BR, McDowell CP, Hallgren M, Meyer JD, Lyons M, Herring MP. Association of efficacy of resistance exercise training with depressive symptoms: meta-analysis and meta-regression analysis of randomized clinical trials. JAMA Psychiatry. 2018;75(6):566–76.

• Gordon BR, McDowell CP, Lyons M, Herring MP. The effects of resistance exercise training on anxiety: a meta-analysis and meta-regression analysis of randomized controlled trials. Sports Med. 2017:1–12.

Line 103: re: “…ST is poor’ – is this the right wording? I suggest rewording this to say something like “population-level engagement in strength training is low”. In addition, to further strengthen the case for the current review, you could compare this to MVPA estimates. See https://doi.org/10.1186/s40798-020-00271-w

Line 106: It would be informative to provide examples of barriers that are common to both aerobic and strength training.

Line 122: Since Phillips (2010) and Winnett (2009) are both narrative reviews and are ~10yrs old, these are not the best references here.

Line 123: List the ‘potential correlates’ that the review by Rhodes et al., (2018) found.

Lines 123-127: Suggest breaking this up into 2-3 sentences. To me, your argument/point gets lost if you deliver this all in one long sentence. Also, I feel the use of BCT could be set up better in this section. You could briefly describe how BCT have been successfully used to positively influences other health-related behaviours (e.g. MVPA, smoking, alcohol, weight loss) if that is indeed the case.

METHODS

Lines 137-157: I found this whole section not that relevant to the paper. For example, readers do not need to know the specifics of the APAB, nor the background of the researchers. I would suggest either deleting completely or significantly reducing this text. If you want to acknowledge the APAB, do so in the acknowledgements section.

Line 183: As noted above, I am curious why you only selected ‘strength training OR resistance training’ as the key search terms to define strength training? The way strength training is defined in the literature is not consistent and may incorporate many different terms: for example: " weight training" "weight lifting" "muscle strengthening" "muscle toning" "weight bearing training" "weight-bearing strengthening". Are you confident that the non-use of these other terms enabled you to capture all strength training interventions in your database search? Also, what about other exercise modes PA that has some strength components – yoga, pilates, Callanetics etc. You will need to provide a reason why these were not included. Please provide a rationale as to why a wider term for strength training wasn’t used in your search strategy.

RESULTS

The results section and tables are provided in a format that is appropriate for a scoping review. I have no major comments.

DISCUSSION

Lines 302-315: Before getting into a discussion on future research, it would be more informative to collectively describe one or two of the key findings of your review, and then how this fits within physical activity promotion. I suggest providing one paragraph on the above, then a separate one talking about future research.

Line 317-328: All this is ok, reading this I was left wondering how does this link to the current studies findings. Incorporate sentences such as “our review to the strength training intervention literature showed that xx and yy….therefore…..”

Line 328: briefly list some of “theories outside of the psychosocial domain may be useful”

Lines 330-353: As per comment above, you should relate this back to what your review showed.

Lines 397-399: add in square brackets for [behaviour] and [practical].

6. PLOS authors have the option to publish the peer review history of their article (what does this mean?). If published, this will include your full peer review and any attached files.

Reviewer #1: No

Reviewer #2: No

---

## [Author Response · Author response to Decision Letter 0]

28 Apr 2021

Please also see attached file in case of formatting errors:

Response to Reviewers

We appreciate the reviewers thoughtful and valuable comments which have greatly helped to strengthen our manuscript. We have responded in detail to all their comments below.

Legend:

Reviewer Comments: Black text

Response to Reviewers: Blue italicized text

*Note: Line numbers are referenced for the track changes copy

PONE-D-20-27748

A scoping review of strength training behaviour change interventions: Future research and practical applications

Journal Requirements:

Thank you for providing templates. We have double checked to ensure our manuscript meets formatting requirements. 

2. We noted that your search was completed in February 2019. Please ensure that your search is updated and any relevant articles published since February 2019 are included in your review.

We have updated the search as of December 2020 and all relevant articles are included in the review.

Thank you, our data availability statement remains the same, we have provided a DOI to access the data from the Open Science Framework.

Additional Editor Comments (if provided):

1. Overall, consider removing the “ST” acronym for strength training as it is not commonly used in the field.

Removed as requested.

2. Please use acronyms consistently throughout the text. For example, line 395 BCT is written out in full in the same paragraph where the acronym appears and the PA acronym appears on line 400 without previously being introduced.

Apologies for the oversight, we have corrected acronyms as requested.

3. For numbers under 10 that appear in text (i.e. not in tables) please write out full. For example, line 249 please change “2” to “two”.

Apologies for the oversight, have corrected as per APA formatting style as requested.

4. Were only 24 strength training interventions described in the literature or only 24 that described using a theory of behaviour change?

Have clarified as:

Line 452: “twenty-seven behavioural interventions targeted towards improving strength training participation met the eligibility criteria.”

5. The search is now almost two years old, have the authors considered an update?

Yes, we have now updated the search as of December 2020 and all relevant articles are included in the review.

6. Why were studies targeting multiple health behaviours simultaneously excluded rather than a sensitivity analysis undertaken?

The reviewer raises an excellent point. However, given the focus of this paper to characterize strength training behaviour interventions as unique from other health behaviour interventions, we felt this specificity was needed. Further, it would be impossible to decipher the behaviour change techniques that were implemented to target one behaviour vs another. Including only interventions that targeted strength training behaviour gives us confidence that the BCTs coded were indeed intended to change strength training behaviour.

7. The description for the grey literature search methods is missing.

We have updated the methods for the grey literature search:

Line 362: Grey literature was searched using the Canadian Agency for Drugs and Technologies in Health Grey Matters Tool, the first 10 pages of Google, and websites of the Obesity Evidence Hub, Fitness Australia, Physiopedia, and National Academy of Sports Medicine.

8. The use of the BCTTv1 training was a nice addition to the methodology.

Thank you.

9. Line 265, please place the examples in parentheses.

Added parentheses

Reviewers' comments:

Reviewer #1: Review comments to authors:

The current study is on a topic of relevance and the idea is innovative. This scoping review has salience towards the planning of behaviour change interventions for strength training. Of greater importance will be the systematic review regarding the efficacy of these interventions which is currently being under-taken (as indicated by the authors in the Discussion section). One option is to combine both papers or submit as companion papers. There is probably enough information focus to warrant both papers.

Thank you, we too agree there is enough information to warrant both papers

The methodological approach for this scoping review appears to have some limitations. 

The reviewer raises a list of excellent points. we will address these point-by-point:

It is not clear how the authors selected the search terms; more detail is warranted. Why were other relevant databases not selected (e.g. SPORTDiscus, Cinahl, Scopus)? We believe that some key studies have not been included in this review.

To increase confidence in our search strategy, we have updated the search to include CINAHL and SPORTDiscus. In doing so, we screened an additional 7990 titles and abstracts since our original screen of 8072 titles and abstracts. Unfortunately, our university does not subscribe to Scopus and were unable to add this database to our search. Of note, no new studies from the original search time frame were identified with the addition of these databases. Three new studies were published since the last search was conducted. We consulted a medical librarian to develop our search terms and a sample strategy is included in the supplementary files. 

Line 297: “A medical librarian was consulted to develop the literature search strategy. Electronic databases were searched for relevant articles. Trial registries and reference lists of selected reviews and included studies were hand-searched. Additionally, we consulted content experts in the field to confirm the final list of included studies. The original search included articles published until February 2019 with an updated search performed December 2020. Embase (1974-present), Medline (1946-present), PsychINFO (1987-present), PubMed (1950-present), CINAHL (1937-present), and SPORTDiscus (1837-present) databases were searched using the following keywords (for a sample search strategy see S2): (1) Terms for interventions included: ‘intervention stud*’ OR ‘program’ OR ‘curriculum’ OR ‘physical education’ OR ‘promotion’ OR ‘initiative’ OR ‘behaviour change’ OR ‘strateg*’, (2) terms for strength training included: strength training OR resistance training OR muscle strengthening”.

It was not entirely clear how the authors selected the included studies, i.e. clarification is required regarding the selection criteria for study selection. On page 8 – it is reported that ‘any study design; was included in the search strategy. However this paper reports on ‘intervention studies’.

Thank you for bringing this to our attention. We have clarified that we only included intervention study designs.

Overall, the manuscript requires major revisions including more focused writing and structuring within each section. Specific examples and suggestions are detailed below.

Thank you for your specific comments, we have addressed each of them below.

Introduction

Overall, the introduction needs to be written in more focused and clear language. For example, the first sentence should be edited to “International guidelines recommend that adults and older adults should engage in muscle strengthening activities at least twice weekly.” 

Edited as suggested.

The sentence starting on line 98 also needs editing, it should read “ST is especially important for older adults, where increased strength helps support continued function and independence”

We have modified the reviewer’s suggestion slightly to include comments made by reviewer #2: 

Line 128: “Strength training is especially important for older adults’ independence and cognitive and physical function [1,2].”

Instead of using the term “ST behaviour change interventions”, which reads quite awkwardly. It is recommended that authors reword this to “ST interventions that use/include/utilizes behaviour change strategies”.

We have changed this term to strength training participation interventions for simplicity and to distinguish this behavioural type of intervention from exercise interventions targeted towards improving strength as a health outcome. 

It is also recommended that when authors discuss the low rates of meeting the ST guidelines, that they compare these to the prevalence of meeting the aerobic guidelines.

Amended to: 

Line 133: “Although the health benefits of strength training are known, population-level engagement in strength training is low with only 1-31% of individuals (varying across countries as well as general and patient populations) meeting strength training guidelines [3–8]. These rates are considerably lower than those meeting the aerobic exercise guidelines (~50%) [9,10].”

On a number of locations, the authors mention ‘strengthening activities’, e.g. line 106. It is recommended that the authors reword this to ‘muscle strengthening’ or ‘muscle strengthening activities’ and use this throughout the manuscript.

Agreed, thank you for clarifying. ‘Strengthening activities’ have been changed to ‘muscle strengthening activities’ throughout the manuscript

The sentence starting on line 110 is awkward and requires rewording. For example, “When compared to aerobic activities (e.g. walking), the perceived knowledge and effort required to effectively perform muscle strengthening activities (e.g. specific techniques, cost of equipment, knowledge of how much ST to perform/ how to progress) can be deterring.” Authors also need to reference this sentence.

Corrected as suggested and references added, thank you.

The sentence starts on line 113 is awkward and needs rewording. The authors need to be really clear what they are referring to i.e. interventions targeting a particular type of exercise. For example, “Given the unique barriers to ST, it may be that interventions targeting ST require different behaviour change strategies than those targeting aerobic exercise or ST combined with other physical activities.”

Corrected as suggested, thank you.

The last paragraph of the introduction is awkward and repetitive in places, please revise.

We have attempted to reorganize and remove information where appropriate to improve the clarity of this paragraph.

Line 204-220: Several reviews have synthesized general physical activity behaviour change interventions across diverse populations [11–13]; however, no synthesis of interventions specifically targeting strength training participation has been conducted. One systematic review summarized the behavioural, demographic, intrapersonal, interpersonal, and environmental factors associated with participating in strength training [14]. This review highlights potential correlates, such as self-efficacy, intention, affective judgements, self-regulation, and subjective norms, that are linked to greater strength training behaviour. However, behaviour change interventions are complex and mode of delivery, providers, intervention dose, setting, behaviour change techniques (BCTs), and the exercise prescription itself may also influence the effectiveness of interventions. Indeed, BCTs, or the ‘active’ ingredients of an intervention [15], have been extensively summarized across health behaviours such as smoking, diet, and physical activity, but have not been examined specifically across strength training interventions [11,13,16]. With so many variables to consider when designing strength training participation interventions, it is important to summarize what has been tested to help set a research agenda for intervention development. The purpose of this scoping review was to map the intervention characteristics, prescription parameters, and BCTs used in interventions to improve strength training participation to date.

Methods

The method needs to be written in more focused language and be more structured. For example:

It is recommended that authors use an appropriate subheading for the first paragraph.

We have revised the subheading to “Methodology” and later in the methods, added the subheading “stakeholder engagement”

Page 6. This paragraph is confusing and wordy, please revise using more focused language.

We have attempted to provide a more concise description of our stakeholder involvement process while trying to maintain key elements of the GRIPP standards for reporting patient involvement (Staniszewska, et al., 2017): 

Line 437-444: “The Patient Engagement In Research framework was used to guide our involvement of patient/healthcare provider partners at relevant stages of the review process to improve the usability of recommendations to end-users [17,18]. Specifically, Arthritis Research Canada’s Arthritis Patient Advisory Board ([APAB] a group of advocates that bring lived experience and patient knowledge to research decision making) was consulted to shape the research question. Two patient/clinician partners (AH, KT) contributed to the interpretation of the findings and development of the paper.”

Page 6. The last sentence of the paragraph should be removed.

This sentence has been removed.

It is also recommended that authors remove the ‘Stage 1’, ‘Stage 2’ etc from each of the subheadings as it is expected that each section follows in a chronological fashion.

The use of stages has been removed.

Page 8. In the study selection. While the authors include the inclusion criteria and exclusion criteria. Authors also state on line 191 “Several strength training interventions have been conducted to date, the inclusion of articles with a measure of ST participation as an outcome was selected to help distinguish behavioural interventions from general ST training interventions”. Please clarify this.

We have clarified:

Line 371: “The inclusion of articles with a measure of strength training participation as an outcome was selected to help distinguish behavioural interventions that are designed to improve strength training participation from interventions that were designed to improve health outcomes.”

On page 9. Authors are using the heading “Charting the data”, please change to heading “Data extraction” as this terminology is used in the abstract. It is important to be consistent with the language used throughout the manuscript.

Corrected as suggested, apologies for the inconsistency.

On page 9, line 215, the authors have written 2.4.1 – please remove.

Removed

On page 9, change the subheading “Collating, summarizing, and reporting the results” to “Analysis”

We have changed this to “data analysis and presentation” as per Joanna Briggs Institute Evidence Synthesis Scoping Review guidelines (Peters MDJ, Godfrey C, McInerney P, Munn Z, Tricco AC, Khalil, H. Chapter 11: Scoping Reviews (2020 version). In: Aromataris E, Munn Z (Editors). JBI Manual for Evidence Synthesis, JBI, 2020. https://doi.org/10.46658/JBIMES-20-12)

Results

On page 10, under “Study characteristics” and “Intervention characteristics”, instead of spelling out all the information that is already presented in Table 1, do a brief summary. Alternatively, remove Table 1 and Table 2 and keep the text.

Thank you for bringing this to our attention. All of the paragraphs within the results section have been amended to avoid duplication of text and table information.

Please include the range of participants i.e. ranging between X to 3500.

Range has been added:

Line 459: “Participant sample sizes ranged from 6 [19] to 3500 participants [20]…”

It is also recommended that the authors edit Table 3. In the first column, add in the reference number as opposed to stating the author and year. In Table 3, the authors mention 1-RM in the ‘Exercise volume’ column and the number of sets in the exercise intensity? 1-RM is a measure of intensity, please go through the table and amend accordingly. In addition, second column under ‘Exercise frequency’, this information is often repeated under the ‘Intervention procedure’ column. Please revise and avoid repeating information.

Thank you for your comment. We elected not to change the Author, Year format to reference numbers in order to avoid confusion for the reader; the references for all included studies are found in supplementary files meaning a given reference in numerical format could mean two different papers whether the reader looks at the main file vs supplementary file. Intensity and volume information have been corrected for Lubans et al. (2012). We have reviewed the remainder of this table for correct classification of information within the exercise volume and exercise intensity columns. Apologies for the confusion re: frequency in two columns. We have tried to better define table titles to reflect their content and although they may share similar frequencies in some instances, that these are different items being captured and not repetitions of content. As such, exercise frequency is now labelled under recommended exercise prescriptions and ‘intervention description’ is now clarified as “any contacts made between interventionists and participants”. 

Discussion

Overall, the discussion requires major revisions, including being written in a more structured and focused language. Some examples listed below:

Page 20, line 302, avoid ‘I’ or ‘We’ statements, please amend to “The present review identified 24 unique interventions for improving ST behaviour”.

‘I’ or ‘we’ statements have been removed throughout. 

In the second paragraph “Opportunity to test theory” the authors are suggesting a number of theories to include, HAPA, SDT and TPB, however the authors do not justify why these ones should be tested? Have they been used in other aerobic and/or aerobic + RT interventions? The authors should remove the sentence on line 321-324 or reword as it doesn’t really fit into the context. All those constructs are part of SCG.

The reviewer makes an excellent observation. We have revised our discussion on theory to focus on i) highlighting that only three theories have been tested in strength training participation interventions, ii) a comparison to the available behaviour change theories and the spectrum of theories used in the general physical activity literature, ii) constructs identified as effective targets for increasing strength training participation, iv) suggesting that specific recommendations for next theories to test are premature, while identifying that theories that explore non-conscious processes or the role of the environment remain untested (lines 702-754) .

The ‘Intervention characteristics: consideration of practical interventions” paragraph is confusing and hard to follow. The authors needs to rethink their discussion points.

The authors agree with your comments and have refocussed our discussion points to address the gaps in understanding: 

1) Exercise prescriptions: The influence of prescription parameters on strength training participation 

2) Mode of delivery: Whether in-person interventions specific to strength training are necessary or whether technology (e.g., online videos, video conferencing, mobile apps) can replace person-to-person contact.

3) Interventionist: The needs and potential for non-exercise specialists to improve strength training participation 

4) Other intervention characteristics that influence affect: Strategies such as variety, group activities, or alternative muscle strengthening activities that may affect the positive or negative experiences of strength training

(Lines 756-924)

Throughout the discussion, the authors could remove the first sentence under each subheading or remove the subheadings, as the first sentence is repetition from the previous paragraph.

The reviewer makes a good point, we have removed opening sentences where appropriate. 

Page 24, line 407, please remove the statement “frequency does not infer effectiveness” unless the authors can reference this. 

Added reference to a meta-analysis of physical activity interventions coded using BCTs that supports this statement. 

Also reword the next sentence to make it more generalised i.e. other BCT’s may be effective (equally, less or more), but we don’t know as these haven’t been tested.

Have reworded to Line 960: “It is possible that other, less frequently studied or even untested BCTs are effective for changing strength training behaviour.”

References

Some references are listed two or more times eg Lubans ,et al 2013. Please go through the reference list and remove duplicates where applicable.

We have reviewed the reference list and removed duplicates, apologies for the oversight.

Reviewer #2: A scoping review of strength training behaviour change interventions: Future research and practical applications

REVIEWER COMMENTS

Overview

Thank you for the opportunity to review this manuscript. This paper addresses an important and currently understudied area in physical activity and public health – the use of behaviour change science in strength training. Overall, the paper is generally well-written and provides some interesting and noteworthy findings on an understudied topic.

Sincere thanks for your time in review and positive feedback. Your review is well-received and especially appreciated for the time you’ve taken to provide constructive feedback and seminal references. 

However, I do have a major concern regarding the search strategy used. For example, reading this, I wondered if the use of only ‘strength training’ OR ‘resistance training’ has resulted in missing some interventions in this area. It should be noted that way strength training is currently defined in the literature is not consistent and may incorporate many different terms: for example: " weight training" "weight lifting" "muscle strengthening" "muscle toning" "weight-bearing training" "weight-bearing strengthening". On this, in my comments below, I have asked the authors to defend the use of this ‘narrow’ search strategy. If they can not, I suggest re-running the search to include a broader search of possible strength training -related terms.

Thank you, we have addressed this comment following your specific recommendations under the ‘METHODS’ section.

I will now provide my comments on each section of the manuscript in order of presentation.

INTRODUCTION

First, I suggest not abbreviating strength training to ‘ST’ throughout the manuscript. I feel that this would enhance the readability of the manuscript.

We have removed the acronym as suggested.

Second, this section could be strengthened with the inclusion of more current/relevant references, as well as some edits to improve readability. A recent review article that provides an overview of this strength training in epidemiology/public health and may be useful to read and incorporate into your introduction: https://doi.org/10.1186/s40798-020-00271-w

Many thanks for your suggestions. We have updated the introduction to include more current/relevant references and attempted to address readability throughout the introduction. Our amendments to address your specific comments are listed below. 

Specific suggestions

Line 93: ‘International guidelines’ is very vague – I suggest that you change this to ‘International physical activity guidelines for public health’

Amended as requested.

Lines 96-98 – many of these references are either not current or not that relevant. I suggest including the following:

Cardiometabolic health:

• Ashton RE, Tew GA, Aning JJ, Gilbert SE, Lewis L, Saxton JM. Effects of short-term, medium-term and long-term resistance exercise training on cardiometabolic health outcomes in adults: systematic review with meta-analysis. Br J Sports Med. 2018:bjsports-2017-098970.

• Lemes ÍR, Ferreira PH, Linares SN, Machado AF, Pastre CM, Netto J. Resistance training reduces systolic blood pressure in metabolic syndrome: a systematic review and meta-analysis of randomised controlled trials. Br J Sports Med. 2016:bjsports-2015-094715

TYPE II DIABETES

• Grontved A, Pan A, Mekary RA, Stampfer M, Willett WC, Manson JE, et al. Muscle-strengthening and conditioning activities and risk of type 2 diabetes: a prospective study in two cohorts of US women. PLoS Med. 2014;11(1):e1001587.

• Grontved A, Rimm EB, Willett WC, Andersen LB, Hu FB. A prospective study of weight training and risk of type 2 diabetes mellitus in men. Arch Intern Med. 2012;172(17):1306–12.

ALL-CAUSE MORTALITY

• Saeidifard F, Medina-Inojosa JR, West CP, Olson TP, Somers VK, Bonikowske AR, et al. The association of resistance training with mortality: a systematic review and meta-analysis. Eur J Prev Cardiol. 2019;2047487319850718.

You should also cite some of the recent reviews on mental health and strength training, as these are very important as well

• Gordon BR, McDowell CP, Hallgren M, Meyer JD, Lyons M, Herring MP. Association of efficacy of resistance exercise training with depressive symptoms: meta-analysis and meta-regression analysis of randomized clinical trials. JAMA Psychiatry. 2018;75(6):566–76.

• Gordon BR, McDowell CP, Lyons M, Herring MP. The effects of resistance exercise training on anxiety: a meta-analysis and meta-regression analysis of randomized controlled trials. Sports Med. 2017:1–12.

We sincerely appreciate the time you’ve taken to list these seminal articles. We have added these to our library and updated the introduction accordingly.

As an example, line 125: “Regular strength training can provide benefits to cardiometabolic health such as lowering blood lipids (Tambalis, Panagiotakos, Kavouras, & Sidossis, 2009), blood pressure (Ashton et al., 2020), risk for Type II diabetes (Grøntved et al., 2014; Grøntved, Rimm, Willett, Andersen, & Hu, 2012), and has demonstrated reductions in anxiety (Gordon, McDowell, Lyons, & Herring, 2017), depressive symptoms (Gordon et al., 2018) and all-cause mortality (Saeidifard et al., 2019; Stamatakis et al., 2018). Strength training is especially important for older adults’ independence and cognitive and physical function (Falck et al., 2019; Fragala et al., 2019).”

Line 103: re: “…ST is poor’ – is this the right wording? I suggest rewording this to say something like “population-level engagement in strength training is low”. In addition, to further strengthen the case for the current review, you could compare this to MVPA estimates. See https://doi.org/10.1186/s40798-020-00271-w

Amended to, Line 133: “Although the health benefits of strength training are known, population-level engagement in strength training is low with only 1-31% of individuals (varying across countries as well as general and patient populations) meeting strength training guidelines [3–8]. These rates are considerably lower than those meeting the aerobic guidelines (~50%) [9].”

Line 106: It would be informative to provide examples of barriers that are common to both aerobic and strength training.

Line 135: Amended to: “These low rates of strength training participation may be attributed to the many barriers that are observed across both aerobic and muscle strengthening activities (e.g., perceived lack of time, self-efficacy, cost; [29,30])”

Line 122: Since Phillips (2010) and Winnett (2009) are both narrative reviews and are ~10yrs old, these are not the best references here.

Agreed, we have updated to the NSCA’s 2019 position statement on resistance training for older adults.

Fragala, M. S., Cadore, E. L., Dorgo, S., Izquierdo, M., Kraemer, W. J., Peterson, M. D., & Ryan, E. D. (2019). Resistance Training for Older Adults: Position Statement From the National Strength and Conditioning Association. Journal of Strength and Conditioning Research, 33(8), 2019–2052. https://doi.org/10.1519/JSC.0000000000003230

Line 123: List the ‘potential correlates’ that the review by Rhodes et al., (2018) found.

Updated, Line 208: “This review highlights potential correlates, such as self-efficacy, intention, affective judgements, self-regulation, subjective norms that may be targeted to change strength training behaviour…”

Lines 123-127: Suggest breaking this up into 2-3 sentences. To me, your argument/point gets lost if you deliver this all in one long sentence. Also, I feel the use of BCT could be set up better in this section. You could briefly describe how BCT have been successfully used to positively influences other health-related behaviours (e.g. MVPA, smoking, alcohol, weight loss) if that is indeed the case.

Thank you for an excellent suggestion, have amended to:

Line 208-216: “This review highlights potential correlates, such as self-efficacy, intention, affective judgements, self-regulation, and subjective norms, that are linked to greater strength training behaviour. However, behaviour change interventions are complex and mode of delivery, providers, intervention dose, setting, behaviour change techniques (BCTs), and the exercise prescription itself may also influence the effectiveness of interventions. Indeed, BCTs, or the ‘active’ ingredients of an intervention [15], have been extensively summarized across health behaviours such as smoking, diet, and physical activity, but have not been examined specifically across strength training interventions [11,13,16]”

METHODS

Lines 137-157: I found this whole section not that relevant to the paper. For example, readers do not need to know the specifics of the APAB, nor the background of the researchers. I would suggest either deleting completely or significantly reducing this text. If you want to acknowledge the APAB, do so in the acknowledgements section.

Thank you for your comment, we agree some of the included text was unnecessary. We have significantly reduced the text but have not removed the stakeholder engagement description. The role of patient/clinician partners in conducting reviews and other research is an important, albeit burgeoning area, and requires adequate reporting:

Staniszewska, S., Brett, J., Simera, I., Seers, K., Mockford, C., Goodlad, S., ... & Tysall, C. (2017). GRIPP2 reporting checklists: tools to improve reporting of patient and public involvement in research. bmj, 358.

We have amended to: Line 438-444: “The Patient Engagement In Research framework was used to guide our involvement of patient/healthcare provider partners at relevant stages of the review process to improve the usability of recommendations to end-users [17,18]. Specifically, Arthritis Research Canada’s Arthritis Patient Advisory Board ([APAB] a group of advocates that bring lived experience and patient knowledge to research decision making) was consulted to shape the research question. Two patient/clinician partners (AH, KT) contributed to the interpretation of the findings and development of the paper.”

Line 183: As noted above, I am curious why you only selected ‘strength training OR resistance training’ as the key search terms to define strength training? The way strength training is defined in the literature is not consistent and may incorporate many different terms: for example: " weight training" "weight lifting" "muscle strengthening" "muscle toning" "weight bearing training" "weight-bearing strengthening". Are you confident that the non-use of these other terms enabled you to capture all strength training interventions in your database search? Also, what about other exercise modes PA that has some strength components – yoga, pilates, Callanetics etc. You will need to provide a reason why these were not included. Please provide a rationale as to why a wider term for strength training wasn’t used in your search strategy.

An excellent comment. The literature search strategy was developed in collaboration with a medical librarian. The term ‘resistance training’ was searched as an ‘exploded’ MESH term which in for example, EMBASE, includes terms such as resistance exercise, resistance exercise training, strength training, and weight bearing exercise.

1. strength training.mp.

2. exp resistance training/

3. resistance training.mp.

4. muscle strengthening.mp.

5. community program.mp. or exp community program/

6. intervention study.mp. or exp intervention study/

7. physical education.mp. or exp physical education/

8. promotion.mp.

9. exp curriculum/ or curriculum.mp.

10. initiative.mp.

11. behaviour change.mp. or exp behavior change/

12. strategy.mp.

13. 1 or 2 or 3 or 4

14. 5 or 6 or 7 or 8 or 9 or 10 or 11 or 12

15. 13 and 14

In response to reviewer #1 and #2’ comments, we updated our search. In the updated search, I ran a sample search in EMBASE to see the impact of including ‘muscle strengthening’ as a search term and search results returned 549 articles compared to 520, a discrepancy of 5%. The search term ‘muscle strengthening’ was added to the updated search using additional databases including CINAHL and Sportdiscus. With the expanded search terms and databases included, a total of 11 988 titles were screened. This expanded search has increased our confidence that all included articles were screened given:

1) The rigour of our search process included co-development of the search strategy with a medical librarian who specializes exercise-related topics, hand searching of reference lists of the included studies, screening conducted in duplicate, and consultation with experts to confirm our final list

2) Unfortunately, the most similar review to ours (Rhodes et al., 2017; BJSM: http://dx.doi.org/10.1136/bjsports-2016-096950) did not specify the number of titles screened for comparison. We did search their reference list to ensure all intervention studies included in their list were included in our review. Further, a previously conducted meta-analysis with similar physical activity intervention inclusion criteria (https://doi.org/10.1016/j.psychsport.2018.01.006) screened 7040 articles in 2018.

Taken together, acknowledging the terms included in MESH headings, our updated search, rigour in the screening process, and the large number of titles screened (11 988) that is comparable or greater to previous reviews gives us confidence that our search found all or nearly all available interventions that meet our inclusion criteria.

The reviewer raises an excellent point re: the inclusion of yoga, Pilates, callisthenics. Again, modelling the search criteria used in Rhodes et al., 2017 paper in BJSM, they too did not include these forms of exercise in their review. While it is possible that strength adaptations can be observed following these forms of exercise, we are hesitant to include them in our review which argues that strength training interventions differ from aerobic interventions given unique barriers such as cost of equipment, fear of becoming too bulky, knowledge of progressive resistance exercise prescription parameters, etc. [14,31–33]. It is unclear whether these forms of exercise would encounter the same barriers. Nonetheless, we acknowledge the validity of the reviewer’s point and have added to the limitations section: 

Line 1035: Further, alternative forms of exercise which may qualify as muscle strengthening such as yoga, calisthenics, and Pilates were not included in the search and may be important to examine in the future.

RESULTS

The results section and tables are provided in a format that is appropriate for a scoping review. I have no major comments.

Thank you.

DISCUSSION

Lines 302-315: Before getting into a discussion on future research, it would be more informative to collectively describe one or two of the key findings of your review, and then how this fits within physical activity promotion. I suggest providing one paragraph on the above, then a separate one talking about future research.

We have restructured the first paragraph as recommended and elaborate upon future directions in the following paragraphs

Line 317-328: All this is ok, reading this I was left wondering how does this link to the current studies findings. Incorporate sentences such as “our review to the strength training intervention literature showed that xx and yy….therefore…..”

The reviewer makes an excellent observation. We have revised our discussion on theory to focus on i) highlighting that only three theories have been tested in strength training participation interventions, ii) a comparison to the available behaviour change theories and the spectrum of theories used in the general physical activity literature, ii) constructs identified as effective targets for increasing strength training participation, iv) suggesting that specific recommendations for next theories to test are premature, while identifying that theories that explore non-conscious processes or the role of the environment remain untested (lines 702-754) .

Line 328: briefly list some of “theories outside of the psychosocial domain may be useful”

We have removed this sentence to help focus the discussion

Lines 330-353: As per comment above, you should relate this back to what your review showed.

Thank you for bringing this to our attention. We have amended our paragraph on exercise prescription to address the findings that prescriptions for sets, reps, intensity, and/or progression were highly variable across studies with no interventions examining the impact of prescription parameters on strength training behaviour.

Lines 397-399: add in square brackets for [behaviour] and [practical].

Amended as suggested.

---

## [Decision Letter · Decision Letter 1]

23 Jun 2021

PONE-D-20-27748R1

A scoping review of interventions to improve strength training participation

PLOS ONE

Dear Dr. Ma,

Thank you for submitting your manuscript to PLOS ONE. After careful consideration, we feel that it has merit but does not fully meet PLOS ONE’s publication criteria as it currently stands. Therefore, we invite you to submit a revised version of the manuscript that addresses the points raised during the review process.

Thank you for addressing the reviewers' concerns. There are a few remaining minor style edits required before acceptance. Please see below for specific comments.

We look forward to receiving your revised manuscript.

Kind regards,

Stephanie Prince Ware, PhD

Academic Editor

PLOS ONE

Journal Requirements:

Additional Editor Comments (if provided):

1. In the abstract, please include the search date ranges.

2. In the abstract, PsycINFO is misspelled.

3. In the abstract, please describe the type of synthesis used.

4. Page 5, there are two "Methods" and "Methodology" headings, please keep just the "Methods" heading.

5. Page 6, line 128: the research question is not needed as it is essentially a rephrasing of the objectives provided in the background.

6. Page 7, line 141: PsycINFO is misspelled.

7. Page 7: Can the authors confirm this is the complete search strategy? Please provide at least 1 full strategy for a sample database in supplemental material.

8. Page 10, line 215: please spell out 5 (i.e. five).

9. Page 10, line 220: please spell out 2 (i.e. two).

10. Table 1, please include age ranges for the populations.

11. Page 12, line 237: please spell out 7 (i.e. seven).

Reviewers' comments:

Reviewer's Responses to Questions

**Comments to the Author**

1. If the authors have adequately addressed your comments raised in a previous round of review and you feel that this manuscript is now acceptable for publication, you may indicate that here to bypass the “Comments to the Author” section, enter your conflict of interest statement in the “Confidential to Editor” section, and submit your "Accept" recommendation.

Reviewer #1: All comments have been addressed

Reviewer #2: All comments have been addressed

2. Is the manuscript technically sound, and do the data support the conclusions?

Reviewer #1: Yes

Reviewer #2: Yes

3. Has the statistical analysis been performed appropriately and rigorously? 

Reviewer #1: N/A

Reviewer #2: Yes

4. Have the authors made all data underlying the findings in their manuscript fully available?

Reviewer #1: Yes

Reviewer #2: Yes

5. Is the manuscript presented in an intelligible fashion and written in standard English?

Reviewer #1: No

Reviewer #2: Yes

6. Review Comments to the Author

Reviewer #1: The authors have adequately addressed the comments.

However, there are a number of places throughout the paper where the grammar, structure and wording could be improved. I suggest some minor editing to this effect.

Reviewer #2: (No Response)

7. PLOS authors have the option to publish the peer review history of their article (what does this mean?). If published, this will include your full peer review and any attached files.

Reviewer #1: No

Reviewer #2: **Yes: **Jason Bennie

---

## [Author Response · Author response to Decision Letter 1]

23 Jul 2021

Response to Reviewers

We appreciate the reviewers thoughtful and valuable comments which have greatly helped to strengthen our manuscript. We have responded in detail to all their comments below.

Legend:

Reviewer Comments: Black text

Response to Reviewers: Blue italicized text

*Note: Line numbers are referenced for the track changes copy

PONE-D-20-27748

A scoping review of strength training behaviour change interventions: Future research and practical applications

PONE-D-20-27748R1

A scoping review of interventions to improve strength training participation

PLOS ONE

Dear Dr. Ma,

Thank you for submitting your manuscript to PLOS ONE. After careful consideration, we feel that it has merit but does not fully meet PLOS ONE’s publication criteria as it currently stands. Therefore, we invite you to submit a revised version of the manuscript that addresses the points raised during the review process.

Thank you for addressing the reviewers' concerns. There are a few remaining minor style edits required before acceptance. Please see below for specific comments.

We look forward to receiving your revised manuscript.

Kind regards,

Stephanie Prince Ware, PhD

Academic Editor

PLOS ONE

RESPONSE: We sincerely thank the reviewers and PLOS One for their time and thought in review. We have outlined our revisions point-by-point as indicated by the text in blue.

Journal Requirements:

RESPONSE: The reference list has been reviewed. 

Additional Editor Comments (if provided):

1. In the abstract, please include the search date ranges.

RESPONSE: Added “(inception-December 2020)”

2. In the abstract, PsycINFO is misspelled.

RESPONSE: Corrected, apologies for the oversight.

3. In the abstract, please describe the type of synthesis used.

RESPONSE: Line 40: Have included “Data were synthesized using descriptive and frequency reporting.”

4. Page 5, there are two "Methods" and "Methodology" headings, please keep just the "Methods" heading.

RESPONSE: Removed the “Methodology” heading.

5. Page 6, line 128: the research question is not needed as it is essentially a rephrasing of the objectives provided in the background.

RESPONSE: The research question has been removed.

6. Page 7, line 141: PsycINFO is misspelled.

RESPONSE: Corrected, apologies for the oversight.

7. Page 7: Can the authors confirm this is the complete search strategy? Please provide at least 1 full strategy for a sample database in supplemental material.

RESPONSE: Yes, we can confirm that the full search strategy is included in supplemental material.

8. Page 10, line 215: please spell out 5 (i.e. five).

RESPONSE: Corrected.

9. Page 10, line 220: please spell out 2 (i.e. two).

RESPONSE: Corrected.

10. Table 1, please include age ranges for the populations.

RESPONSE: Age ranges have been added.

11. Page 12, line 237: please spell out 7 (i.e. seven).

RESPONSE: Corrected.

Reviewers' comments:

Reviewer's Responses to Questions

Comments to the Author

1. If the authors have adequately addressed your comments raised in a previous round of review and you feel that this manuscript is now acceptable for publication, you may indicate that here to bypass the “Comments to the Author” section, enter your conflict of interest statement in the “Confidential to Editor” section, and submit your "Accept" recommendation.

Reviewer #1: All comments have been addressed

Reviewer #2: All comments have been addressed

2. Is the manuscript technically sound, and do the data support the conclusions?

Reviewer #1: Yes

Reviewer #2: Yes

3. Has the statistical analysis been performed appropriately and rigorously? 

Reviewer #1: N/A

Reviewer #2: Yes

4. Have the authors made all data underlying the findings in their manuscript fully available?

Reviewer #1: Yes

Reviewer #2: Yes

5. Is the manuscript presented in an intelligible fashion and written in standard English?

Reviewer #1: No

Reviewer #2: Yes

6. Review Comments to the Author

Reviewer #1: The authors have adequately addressed the comments.

However, there are a number of places throughout the paper where the grammar, structure and wording could be improved. I suggest some minor editing to this effect.

RESPONSE: We thank the reviewer for their suggestions in this and the previous revision. To address issues related to grammar, structure, and wording, we had a professional editor review and make such changes. Changes are highlighted using track changes throughout. 

Reviewer #2: (No Response)

7. PLOS authors have the option to publish the peer review history of their article (what does this mean?). If published, this will include your full peer review and any attached files.

Do you want your identity to be public for this peer review? For information about this choice, including consent withdrawal, please see our Privacy Policy.

Reviewer #1: No

Reviewer #2: Yes: Jason Bennie

Confirming that our Figures have been uploaded and modified using the suggested PACE tool.

---

## [Decision Letter · Decision Letter 2]

10 Jan 2022

PONE-D-20-27748R2A scoping review of interventions to improve strength training participationPLOS ONE

Dear Dr. Ma,

Thank you for submitting your manuscript to PLOS ONE. After careful consideration, we feel that it has merit but does not fully meet PLOS ONE’s publication criteria as it currently stands. Therefore, we invite you to submit a revised version of the manuscript that addresses the point raised below.

You cite a PROSPERO protocol (CRD42019120251) but PROSPERO does not register scoping reviews. The project was originally registered as a standard systematic review, therefore please add to your paper an explicit mention of this and explain why the study began as a standard systematic review and then was converted to a scoping review. 

We look forward to receiving your revised manuscript.

Kind regards,

Lisa Susan Wieland

Academic Editor

PLOS ONE

Journal Requirements:

Reviewers' comments:

Reviewer's Responses to Questions

**Comments to the Author**

1. If the authors have adequately addressed your comments raised in a previous round of review and you feel that this manuscript is now acceptable for publication, you may indicate that here to bypass the “Comments to the Author” section, enter your conflict of interest statement in the “Confidential to Editor” section, and submit your "Accept" recommendation.

Reviewer #1: All comments have been addressed

2. Is the manuscript technically sound, and do the data support the conclusions?

Reviewer #1: Yes

3. Has the statistical analysis been performed appropriately and rigorously? 

Reviewer #1: Yes

4. Have the authors made all data underlying the findings in their manuscript fully available?

Reviewer #1: Yes

5. Is the manuscript presented in an intelligible fashion and written in standard English?

Reviewer #1: Yes

6. Review Comments to the Author

Reviewer #1: suggest to accept for publication ..........................................................................................................................................................................................................................................................................................

7. PLOS authors have the option to publish the peer review history of their article (what does this mean?). If published, this will include your full peer review and any attached files.

Reviewer #1: **Yes: **Ronald Plotnikoff

---

## [Author Response · Author response to Decision Letter 2]

12 Jan 2022

PONE-D-20-27748

A scoping review of strength training behaviour change interventions: Future research and practical applications

We appreciate the reviewers’ thoughtful and valuable comments throughout the review process. Their feedback has greatly helped to strengthen our manuscript. We have responded in detail to the final comment from the editor below.

Reviewer: You cite a PROSPERO protocol (CRD42019120251) but PROSPERO does not register scoping reviews. The project was originally registered as a standard systematic review, therefore please add to your paper an explicit mention of this and explain why the study began as a standard systematic review and then was converted to a scoping review. 

Response: LINE 423-431: “Second, this review was originally registered with PROSPERO (CRD42019120251) as a systematic review with two research questions. Question one (“what are the effects of strength training participation interventions?”) will be addressed in a separate systematic review. Question two in the registered protocol was defined as “what BCTs, theories, and modes of delivery are used in strength training interventions currently?’. Given the nascency and heterogeneity of the literature, we changed to a scoping review methodology for this specific question. Specifically, it was more appropriate to map the current state of the literature and suggest a research agenda that addresses current gaps than to assess the effectiveness of these intervention components, therefore warranting a scoping review methodology.”

---

## [Editor Report · Decision Letter 3]

18 Jan 2022

A scoping review of interventions to improve strength training participation

PONE-D-20-27748R3

Dear Dr. Ma,

We’re pleased to inform you that your manuscript has been judged scientifically suitable for publication and will be formally accepted for publication once it meets all outstanding technical requirements.

Kind regards,

Lisa Susan Wieland

Academic Editor

PLOS ONE

Additional Editor Comments (optional):

Thank you for clarifying the elements of your original review questions in the PROSPERO registration and their relation to the scoping review you have conducted.
---

## [Editor Report · Acceptance letter]

24 Jan 2022

PONE-D-20-27748R3 

A scoping review of interventions to improve strength training participation 

Dear Dr. Ma:

I'm pleased to inform you that your manuscript has been deemed suitable for publication in PLOS ONE. Congratulations! Your manuscript is now with our production department. 

Kind regards, 

on behalf of

Dr. Lisa Susan Wieland 

Academic Editor

PLOS ONE